# miRGTF-net: Integrative miRNA-gene-TF network analysis reveals key drivers of breast cancer recurrence

Stepan Nersisyan[1]*, Alexei Galatenko[2,3], Vladimir Galatenko[2¤], Maxim Shkurnikov[4], Alexander Tonevitsky[1]

**1** Faculty of Biology and Biotechnology, HSE University, Moscow, Russia, **2** Faculty of Mechanics and Mathematics, Lomonosov Moscow State University, Moscow, Russia, **3** Moscow Center for Fundamental and Applied Mathematics, Moscow, Russia, **4** P.A. Hertsen Moscow Oncology Research Center, Branch of National Medical Research Radiological Center, Ministry of Health of the Russian Federation, Moscow, Russia

¤ Current address: Evotec International GmbH, Göttingen, Germany
* snersisyan@hse.ru

## Abstract

Analysis of regulatory networks is a powerful framework for identification and quantification of intracellular interactions. We introduce miRGTF-net, a novel tool for construction of miRNA-gene-TF networks. We consider multiple transcriptional and post-transcriptional interaction types, including regulation of gene and miRNA expression by transcription factors, gene silencing by miRNAs, and co-expression of host genes with their intronic miRNAs. The underlying algorithm uses information on experimentally validated interactions as well as integrative miRNA/mRNA expression profiles in a given set of samples. The latter ensures simultaneous tissue-specificity and biological validity of interactions. We applied miRGTF-net to paired miRNA/mRNA-sequencing data of breast cancer samples from The Cancer Genome Atlas (TCGA). Together with topological analysis of the constructed network we showed that considered players can form reliable prognostic gene signatures for ER-positive breast cancer. A number of signatures demonstrated remarkably high accuracy on transcriptomic data obtained by both microarrays and RNA sequencing from several independent patient cohorts. Furthermore, an essential part of prognostic genes were identified as direct targets of transcription factor E2F1. The putative interplay between estrogen receptor alpha and E2F1 was suggested as a potential recurrence factor in patients treated with tamoxifen. Source codes of miRGTF-net are available at GitHub (https://github.com/s-a-nersisyan/miRGTF-net).

## Introduction

A microRNA (miRNA) is a short (usually 22 nt long) non-coding RNA which acts as a post-transcriptional gene expression regulator in human cells [1]. Namely, a miRNA incorporated into the RNA-induced silencing complex (RISC) binds the 3′-UTR of the target mRNA which

---

**Data Availability Statement:** All relevant data are within the manuscript and its Supporting information files. Source codes have been made

available on GitHub (https://github.com/s-a-nersisyan/miRGTF-net).

**Funding:** The research was performed within the framework of the Basic Research Program at HSE University.

**Competing interests:** The authors have declared that no competing interests exist.

results in translation repression and/or mRNA degradation [2]. Multiple reports show a crucial role of miRNAs in various pathologies including breast [3–5], lung [6, 7], colon [8, 9] and other cancers [10]. Aside from cancer, miRNAs also contribute to pathogenesis of neurological [11, 12] and cardiovascular [13, 14] diseases as well as several viral infections [15, 16]. Other studies have suggested that miRNAs can also participate in intercellular communication [17, 18].

Network analysis is an efficient strategy for exploration of interplay between miRNAs and genes. In this framework nodes of a network are usually associated with genes and/or miRNAs, while directed edges symbolize interactions between corresponding nodes. Some edges may be endowed with weights that represent certain statistical properties of the interaction (e.g. correlation) [19, 20]. The key aspect in network construction is a set of interaction types covered. A lot of methods consider only classical RISC-mediated interactions between miRNAs and target genes [21–23], while other reports also consider the reciprocal action represented by co-expression of host genes and their intronic miRNAs [24, 25]. Consideration of transcription factors (TFs) brings a new dimension to miRNA-gene networks since TFs can regulate both miRNAs and genes adding gene-gene and more gene-miRNA interactions. For example, construction and analysis of miRNA-TF-gene networks allowed researchers to identify gene-markers in uterine [26] and colorectal cancer [27] and putative drivers of human cleft lip [28].

Various strategies are utilized for network construction. One of possible approaches consists in building networks composed of interactions available in known databases (see, e.g., [29–31]), while other methods lean on co-expression analysis consisting in prediction of interactions based on expression profile in a set of samples [32–34]. Both methods have some limitations. Database-level analysis usually either lacks tissue-specificity (e.g., sequence-based miRNA target prediction) or the number of interactions substantially drops when limited to one particular tissue/cell type (e.g., databases of direct miRNA-mRNA interactions verified by luciferase reporter assays). Correlative studies can lead to a large false-positives rate, since high correlation of expression values does not guarantee actual interaction between two entries. For example, negative correlation of non-interacting miRNA and gene can be due to TF promoting expression of miRNA and inhibiting expression of gene.

In this paper we introduce miRGTF-net, an original tool for constructing miRNA-gene-TF regulatory networks. This tool considers all mentioned transcriptional and post-transcriptional interactions of miRNAs, genes and TFs. Aside from multiple interaction types we utilize a hybrid two-step strategy for edge construction which involves both database-level information and integrative miRNA/gene expression profile in a set of given samples. As a result, miRGTF-net allows one to identify and quantify known interactions specific for analyzed cells or tissues.

The proposed method was used to construct a breast cancer interaction network on the basis of paired miRNA and mRNA sequencing data from The Cancer Genome Atlas Breast Invasive Carcinoma (TCGA-BRCA) project [35]. Apart from topological and connectivity analysis we hypothesized that the most "active" nodes from the network constructed can form reliable prognostic signatures since molecular-level differences in tumors derived from patients with and without recurrence might be due to broken intracellular interactions. To test this hypothesis we established a support vector machine (SVM) classification pipeline for construction of gene signatures separating ER-positive breast cancer patients with and without 5-year recurrence. The pipeline was run against publicly available Affymetrix microarray data from five independent patients cohorts. Then, we explored mutual arrangement of prognostic genes in the constructed network. To test robustness of constructed gene signatures, we measured their performance on patients with expression data profiled by other platforms including the mentioned RNA-seq data from TCGA-BRCA.

## Results

### Overview of the network construction method

We consider a directed graph which nodes are associated with genes and miRNAs, with nodes *A*, *B* connected by a directed edge in the following cases:

- *A* is a gene encoding a transcription factor and *B* is a target gene for *A*;

- *A* is a gene encoding a transcription factor and *B* is a target miRNA for *A*;

- *A* is a miRNA and *B* is a target gene for *A*;

- *A* is a gene which hosts an intronic miRNA *B* with sense orientation.

By default, information about high-confidence interactions is derived from publicly available databases.

After constructing a database-level network we perform quantification of interaction powers using paired miRNA/mRNA expression profiles in a set of samples. Namely, we assign an interaction score to each network edge and incoming score to each node. High absolute value of interaction score is associated with strong linear dependence between two connected nodes in the considered samples, while high value of incoming score means strong linear dependence of the selected node on its direct regulators. The equality of scores to zero means a complete absence of corresponding dependence. The method of quantification is independent of the expression profiling platform and can be evaluated both on RNA-seq and microarray data. However, different techniques can lead to significantly different results, especially for miRNA quantification. In particular, miRNA expression data obtained by hybridization microarrays suffers from high false positive rate (i.e. detection of miRNAs which are actually absent in analyzed samples), and sequencing should be preferred whenever possible [36, 37].

In order to select the most powerful interactions we apply a three-step procedure. First, we select all nodes with significant incoming score together with their incoming edges. After that, we select all edges with significant interaction scores. Finally, we remove isolated nodes from the resulting network. Thus, the resulting network consists of nodes which have significant influence on some other nodes and/or are significantly regulated by some other node.

The workflow of network construction is presented in Fig 1, details are outlined in Methods.

The miRGTF-net is implemented as a user-friendly standalone Python script. The method is not limited just to databases covered in the present manuscript. Specifically, a user can provide an arbitrary set of known interactions. Input of the tool also includes user-specified miRNA/gene expression tables. All parameters and thresholds used by the algorithm can be varied. Output of miRGTF-net consists of reports on network topology and connected components as well as full network files which can be opened by the majority of graph manipulation and analysis tools. Since TCGA contains gene and miRNA expression data for various cancer types, we also prepared instructions on importing and pre-processing data from TCGA. In addition, we pre-included several databases for running miRGTF-net on mouse miRNA/mRNA expression data.

### Breast cancer interaction network

We applied the method proposed to the mRNA/miRNA sequencing data derived from the TCGA-BRCA ER-positive samples subset. The constructed network had 371 nodes, including 141 genes encoding transcription factors, 171 other genes and 59 miRNAs. The highest number of interactions was observed for transcription factors regulating genes (371 entries) and

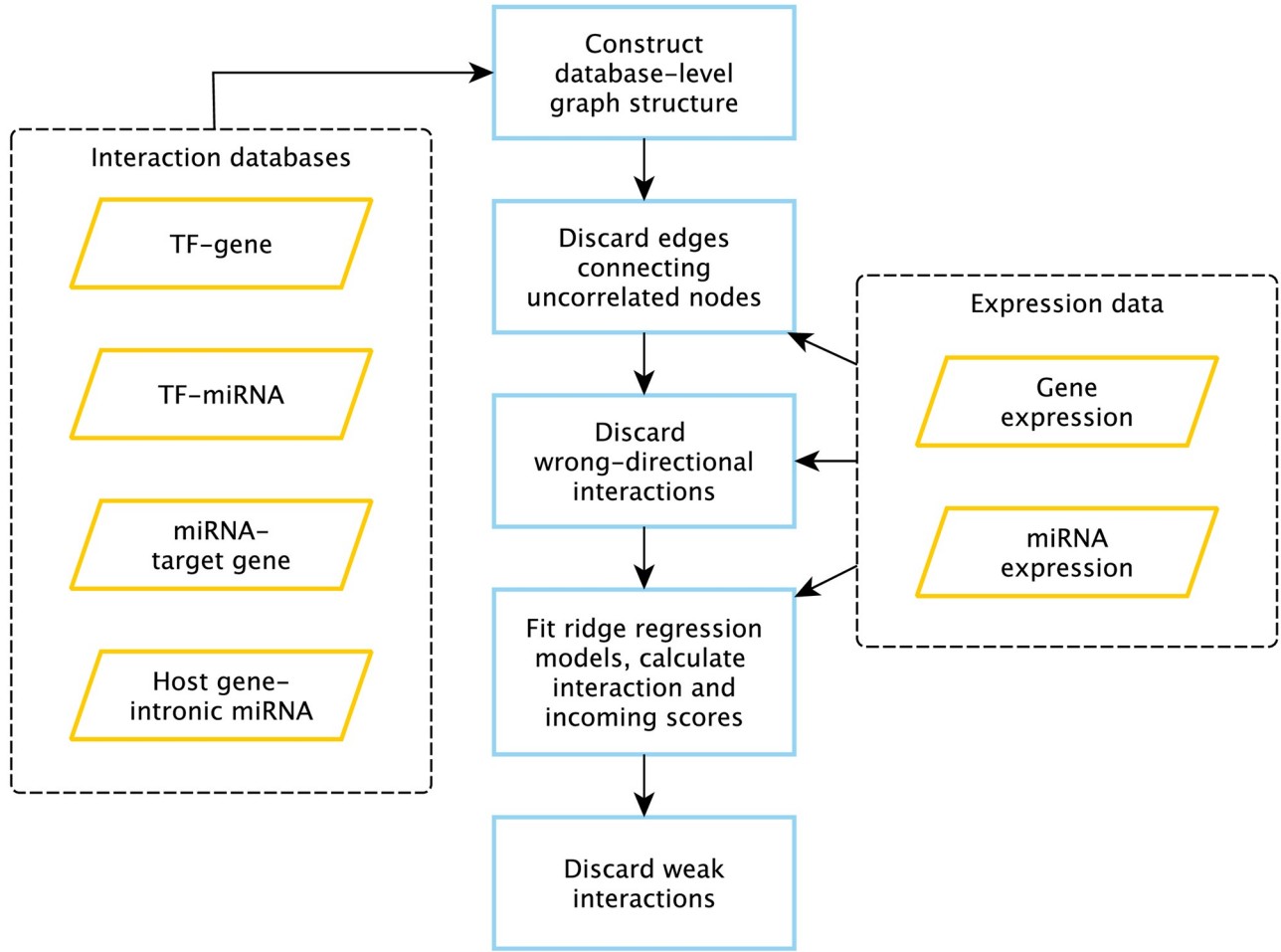

**Fig 1. Workflow of miRGTF-net.** Blue-framed rectangles represent processes, orange-shaped parallelograms represent input data.

miRNAs (73 entries), and miRNAs targeting genes (69 entries), while only 11 edges corresponded to genes hosting intronic sense miRNAs. Weakly connected components of the network (i.e. when all edges are considered undirected) included one large 336-element component, thirteen 2-element components and three 3-element components. It turned out that 9 out of 19 interactions in two- or three-element components represented "isolated" co-transcription of a host gene and an intronic miRNA, while other 10 interactions were interactions between TFs and target genes. For further processing we selected only the large component which is illustrated in Fig 2.

For each network node we calculated in- and out-degrees to represent the most regulated and regulating nodes. About 79% of nodes had at least one incoming edge, while 44% of nodes had one or more outgoing edges. In order to select nodes with significantly high in- or out-degrees we calculated the threshold using random graphs generated by the $G(n, M)$ Erdős–Rényi model [38]; the probability of observing a node with in-/out-degree greater than 8 in a random graph with 336 nodes and 504 edges was equal to 0.0083. Interestingly, not a single node in-degree passed this threshold while there were 13 TFs with such a high out-degree (S1 Table). Three of them, namely, E2F1, SP1 and SPI1, showed especially high out-degrees: 38, 38 and 21, respectively.

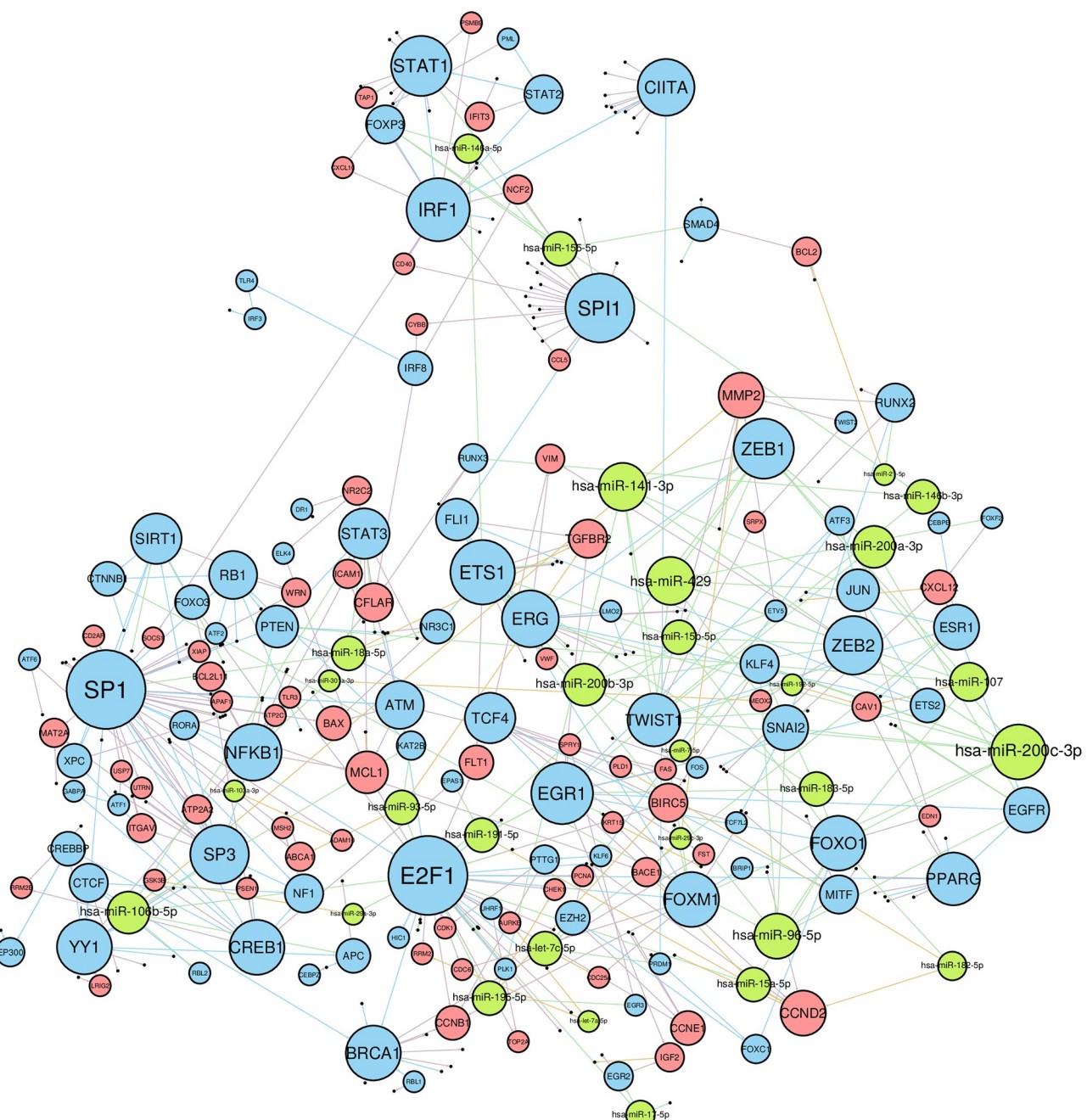

**Fig 2. The main component of the breast cancer interaction network.** Blue nodes represent TFs, red nodes represent genes, green nodes represent miRNAs. The higher node degree is, the bigger is the node size. Edges are colored according to node types they connect.

Functional annotation analysis of 286 network genes revealed 218 significantly enriched terms (S2 Table). As expected, the most enriched terms were directly connected to transcription regulation activity. Cancer and metastasis related terms were also highly represented: 86 genes were related to cell proliferation, 74 to apoptosis, 51 to cell cycle, 45 to DNA damage response, 33 to hypoxia response, while 21 genes were identified as cell adhesion molecules (178 genes in total). Eleven genes were found to share four of these categories, including

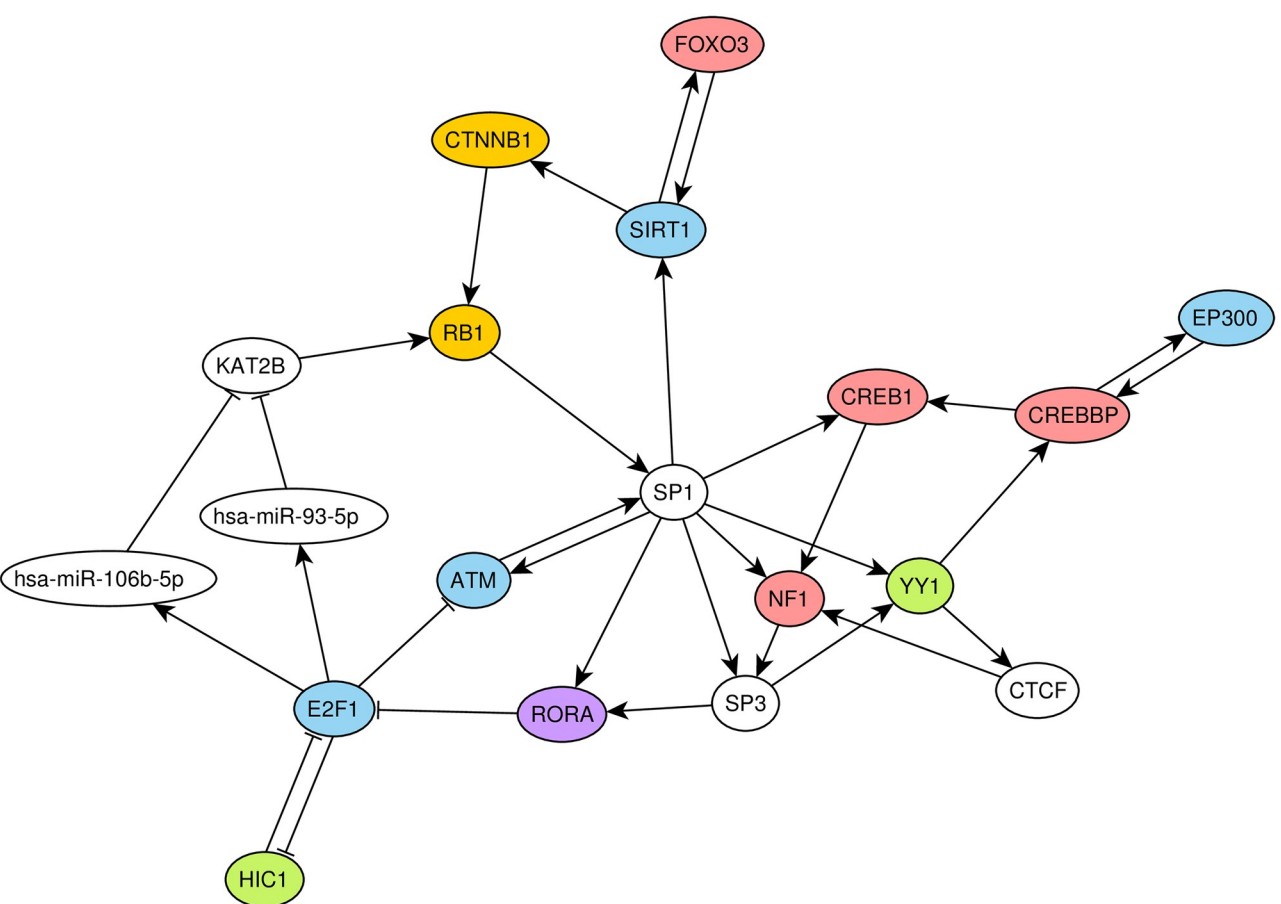

**Fig 3. The first major strongly connected component.** Nodes corresponding to apoptosis, hypoxia and DNA damage responses are colored in blue, nodes corresponding to apoptosis and hypoxia response are colored in red, while nodes associated only to apoptosis, hypoxia response and DNA damage response are colored in orange, purple and green, respectively. The arrows and T-shaped lines signify activation and repression, respectively.

aforementioned E2F1 which contributed to apoptosis, cell cycle, response to DNA damage and hypoxia.

The main component of the constructed network had five non-trivial strongly connected components (i.e. components in which any node can be reached from any other via a directed path), including three 2-element loops (CIITA—IRF1, STAT1—STAT2 and SMAD4—hsa-miR-155-5p) and two major components composed of 19 and 20 nodes. Only three edges connected nodes from two major components, namely, E2F1 regulated transcription of FOXM1 and hsa-miR-15b-5p, while CTNNB1 regulated hsa-miR-96-5p. In the first major component (Fig 3) there were two clustered miRNAs (hsa-miR-93-5p and hsa-miR-106b-5p) and 17 TFs, including ones with the highest out-degrees (E2F1, SP1). This component was enriched by genes associated with cancer. Specifically, the component contained 10 genes related to apoptosis (out of 74 apoptosis-related genes of the network, hypergeometric test $p = 0.0031$), 6 genes associated with DNA damage response (out of 45 genes of the network, $p = 0.0346$), and 9 genes associated with hypoxia response (out of 33 genes, $p = 1.57 \times 10^{-5}$).

The second component was significantly enriched by miRNAs (10 out of 20 nodes, hypergeometric test $p = 0.00013$). The majority of these miRNAs clustered together. Namely, five of them belonged to the mir-200 family (hsa-miR-200a-3p, hsa-miR-200b-3p, hsa-miR-200c-3p, hsa-miR-429, hsa-miR-141-3p), two were from mir-183 family (hsa-miR-183-5p, hsa-miR-96-

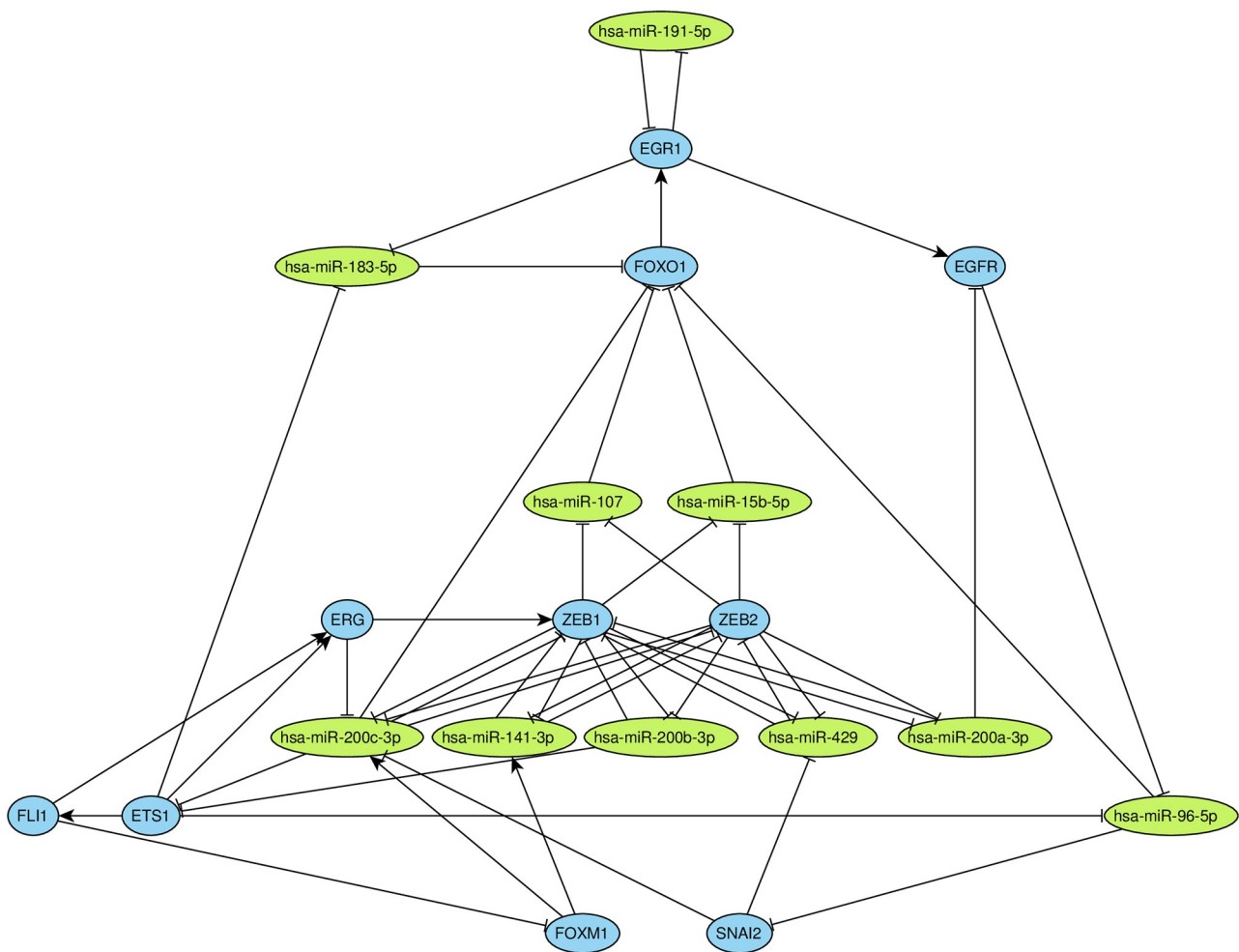

**Fig 4. The second major strongly connected component.** TFs are colored in blue, miRNAs are colored in green. The arrows and T-shaped lines signify activation and repression, respectively.

5p), while the three remaining miRNAs (hsa-miR-15b-5p, hsa-miR-107, hsa-miR-191-5p) were unclustered. The other half of the component was composed of TFs (Fig 4). Interestingly, this component was not statistically significantly enriched by genes from cancer- and metastasis-related categories. However, there is a number of reports suggesting that double-negative feedback loop between mir-200 family and E-cadherin transcriptional repressors ZEB1 and ZEB2 is crucial for metastasis formation, epithelial-mesenchymal transition and invasion (see Discussion for a detailed survey).

## Genes from the network can form highly reliable prognostic signatures

The constructed network consisted of nodes having significant interaction activity at intracellular level. Motivated by the idea that nodes with strong activity can be involved in a high number of corrupted interactions, we hypothesized that these players can form prognostic signatures. In order to reveal such prognostic potential, we developed a classification pipeline to predict 5-year recurrence of ER-positive breast cancer based on expression levels of genes from the network in a primary tumor. The pipeline was tested on publicly available transcriptomic data generated using the Affymetrix Human Genome U133A platform. Since miRNA

expression data was not available in the datasets, we excluded nodes associated with miRNAs from the downstream analysis. To assess a role of the interaction network the pipeline was also launched starting from the list of all genes.

Execution of the pipeline starting from the genes comprising the network resulted in 134 classifiers which passed the filtration thresholds (S3 Table). For certain combination lengths constructed classifiers demonstrated notably high performance on the validation set too. Namely, for combination lengths $k$ = 5, 6, 8, 10 more than 85% of classifiers passed the 0.65 threshold on AUC, sensitivity and specificity (S4 Table). Interestingly, AUC values (calculated on the validation set) were high for all classifiers which passed the filtration: the median AUC calculated for 134 classifiers was equal to 0.77, 95% confidence interval (CI): 0.73-0.79. Thus, the main challenge consisted in identification of a decision threshold suitable for all datasets simultaneously.

The pipeline was also executed starting from the list of all genes. In this case, in contrast to the previous one, only short gene signatures ($k$ = 2, 3, 4) had acceptable performance on the validation set (S4 Table). In general, quality measures were lower for these classifiers compared to the ones composed of genes from the network. In particular, application of Mann-Whitney $U$-test to AUCs on the validation set resulted in $p$-value equal to $3.0 \times 10^{-18}$, indicating statistically significant dominance of AUCs produced by network-based classifiers.

## Prognostic signatures composed of genes from the network demonstrate high compatibility between different gene expression profiling platforms

To examine whether constructed signatures can provide reliable quality on other gene expression profiling platforms, we considered additional samples analyzed by Affymetrix Human Genome U133 Plus 2.0 (a newer version of Affymetrix Human Genome U133A) and RNA sequencing. Despite overall technological similarity of two microarray platforms, they produce significantly different results when analyzing the same sample [39, 40]. Due to fundamental differences between RNA-seq and microarray techniques, expression values obtained by these platforms also demonstrate systematic inconsistency [41, 42].

As a result of applying additional filtration (see subsection Classifier construction pipeline of Methods for details) the pipeline output was reduced to two classifiers composed of genes from the network. The first signature contained five genes: ABCA1, FOS, FOXM1, KIF2C and TMPO. Average AUC, sensitivity and specificity of the classifier on the TCGA dataset were equal to 0.68, 0.64 and 0.6, respectively. At the same time, performance of combination was high on the microarray validation set: AUC, sensitivity and specificity were equal to 0.73, 0.67 and 0.71, respectively. The second combination was composed of eight genes (CDK1, FOXM1, LRIG2, MSH2, PLK1, RACGAP1, RRM2 and TMPO) and demonstrated significantly better performance: average cross-validated AUC, sensitivity and specificity on the TCGA dataset were equal to 0.72, 0.61 and 0.73, while the same metrics were equal to 0.78, 0.67 and 0.71 on the microarray validation set, respectively. The ROC curve for the validation set is presented on Fig 5A. Finally, we performed survival analysis for all patients within the microarray validation set, which resulted in a clear separation of high- and low-risk groups (logrank test $p$ = 0.0015; Fig 5B). Joint distribution of eight genes expression in all used datasets is available in S1 Fig.

In contrast, not a single classifier passed additional microarray and RNA-seq filtration when the pipeline was executed on the list of all genes. Thus, network-based classifiers showed high compatibility between gene expression data obtained by three different platforms. The full information including accuracy metrics on training and filtration sets is available in S3 Table.

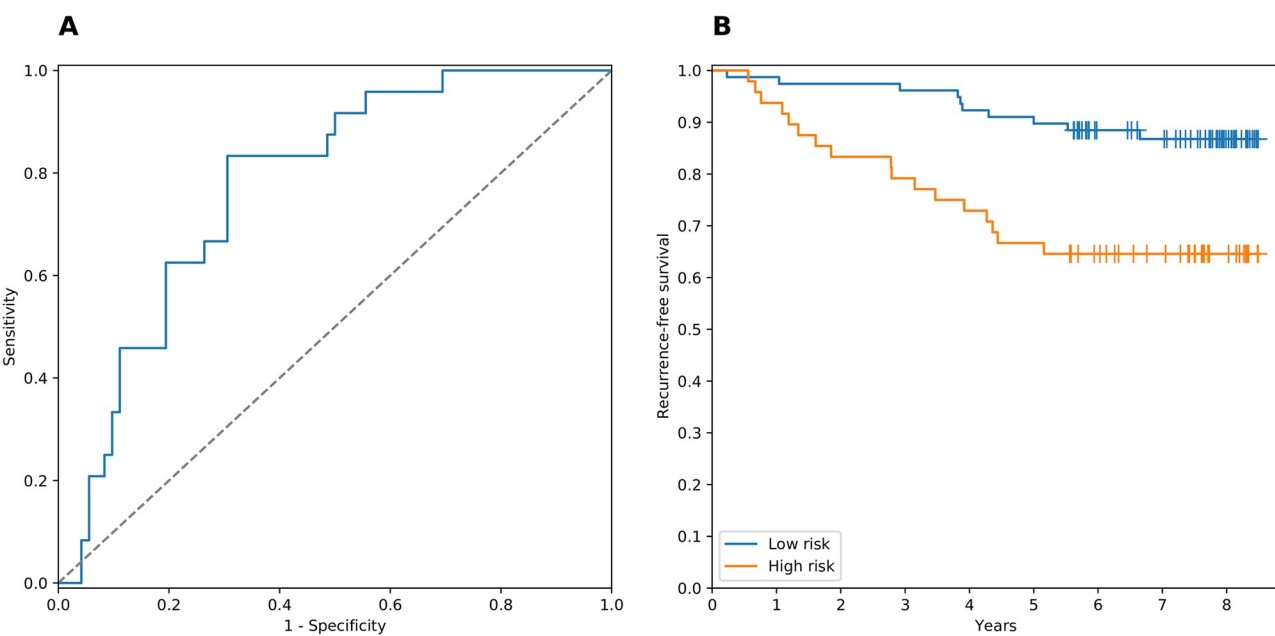

**Fig 5. Performance of the eight-gene signature on the validation set (GSE1456).** (**A**) ROC curve (AUC = 0.78). (**B**) Kaplan-Meier plot (logrank test $p = 0.0015$).

## Activity of E2F1 is crucial for breast cancer recurrence

The signatures generated by the pipeline starting from the network genes contained 49 unique genes. Fourteen of them were present in a significant fraction of combinations (binomial test $p < 0.05$), most of them were involved in cancer-related cellular processes (S5 Table). To explore a potential interplay between these genes we considered a subnetwork composed of them and nodes from their neighborhood (i.e. nodes directly connected with these genes). The resulting subnetwork contained 36 nodes and was weakly connected (Fig 6). Noticeably, seven of considered genes (ATAD2, CCNB1, FOXM1, KIF2C, PLK1, RACGAP1 and RRM2) were directly activated by the E2F1 transcription factor. In particular, this overlap was statistically significant with respect to E2F1 out-degree (hypergeometric test $p = 0.00073$). Moreover, E2F1 was significantly overexpressed in primary tumors of patients with recurrence ($p = 0.021$ in RNA-seq and $p = 0.0017$ in microarray datasets). All seven E2F1 target genes identified above were also up-regulated in the recurrence group (S6 Table).

Patients from analyzed cohorts were treated with tamoxifen, which blocks activity of estrogen receptor alpha encoded by ESR1 gene. As a transcription factor, ESR1 was shown to directly enhance E2F1 expression [43, 44]. While ESR1 and E2F1 were not correlated in the recurrence-free group (Spearman correlation $r = 0.086$, $p = 0.34$), statistically significant positive correlation was observed in tumors of patients with recurrence ($r = 0.22$, $p = 0.013$). Interestingly, ESR1 and E2F1 transcription factors had multiple common targets. Namely, with the use of literature-curated TRRUST database we identified overall fifteen common genes in 76- and 134-element sets of genes regulated by ESR1 and E2F1, respectively (hypergeometric test $p = 1.28 \times 10^{-6}$). This intersection included proteins crucial for cancer pathogenesis such as p53 and Myc transcriptions factors, and apoptosis regulator Bcl-2. The obtained results highlight the potential role of ESR1-E2F1 interplay in the mechanism underlying breast cancer recurrence.

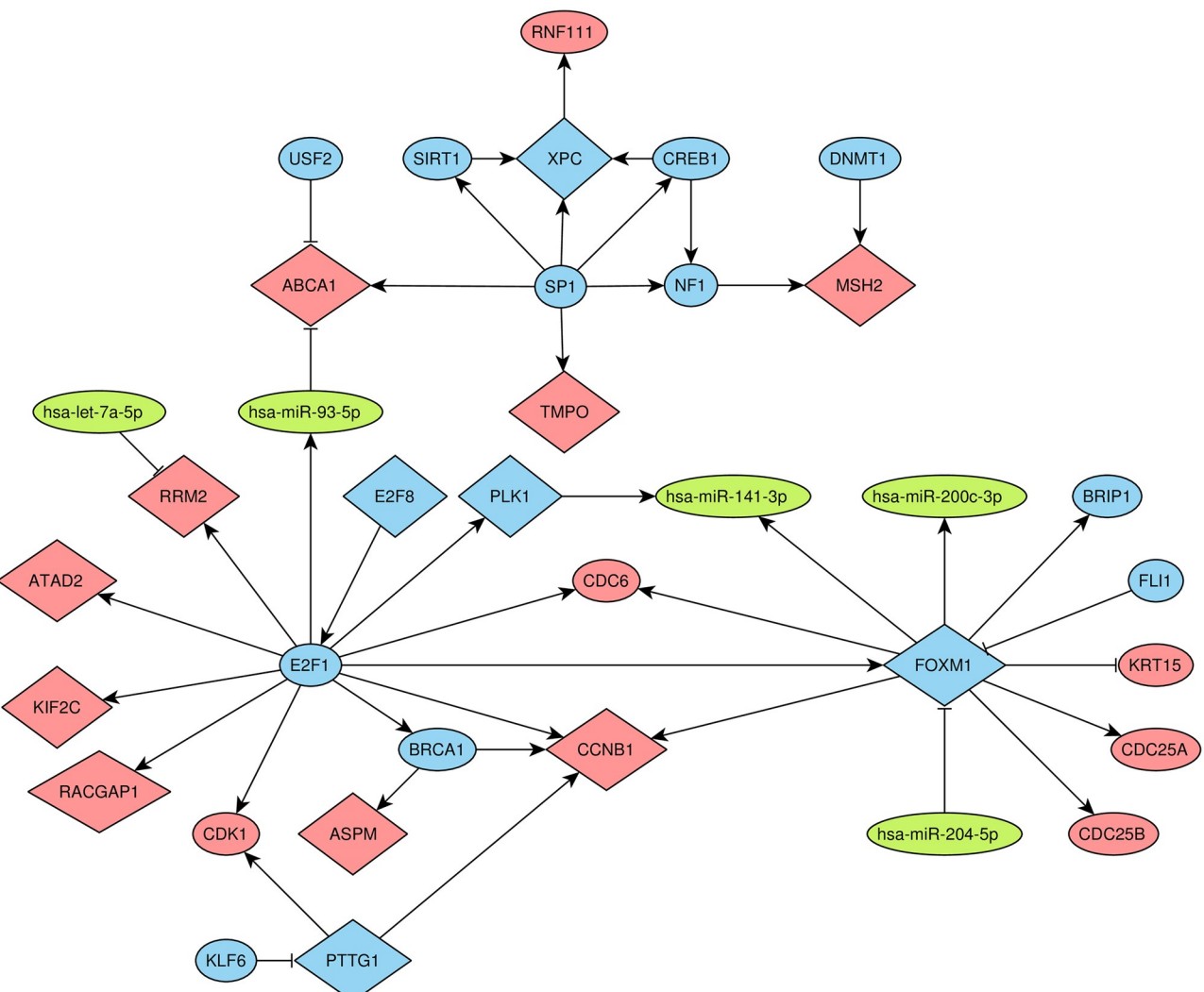

**Fig 6. Neighborhood of genes overrepresented in prognostic signatures.** Diamond-shaped nodes correspond to fourteen overrepresented genes. Blue nodes correspond to TFs, red nodes correspond to genes, green nodes correspond to miRNAs. The arrows and T-shaped lines signify activation and repression, respectively.

## Discussion

The proposed method allows one to construct miRNA-gene-TF interaction networks combining both database-level and integrative miRNA/gene expression data. The advantage of using expression data consists in the ability of quantifying interactions which leads to decreased false positives rate since database interactions can be absent in one or another tissue/cell type. For example, results obtained by application of the proposed method to breast cancer sequencing data suggest an insignificant role of interactions between intronic miRNAs and their host genes, while there is a number of reports which automatically consider all possible interactions of this type [24, 45, 46]: only 11 out of 523 edges (2.1%) were associated with host genes and intronic miRNAs, and the majority of them were isolated. It should be noted that an increasing number of reports highlight the fact that the majority of human miRNAs have independent transcription start sites thus being not co-expressed with their host genes [25, 47].

On the other hand, the need of miRNA and gene expression data is the main limitation for the method. Since some nodes of the network were regulated by tens of other nodes, tens or hundreds of samples are required to fit the regression model appropriately. One possible way to overcome this issue and make the method applicable for smaller amounts of data is to use sparse regression techniques such as lasso [48]. However, $L_1$-regularized models suffer from instability with respect to the input data perturbations when predictor variables are significantly correlated (which is inevitable when dealing with miRNA/gene expression data): only one variable from the correlated group is randomly chosen [49] in contrast to ridge regression which puts equal weights to correlated variables thus solving the multicollinearity problem [50]. Several sophisticated methods such as Sparse-Group Lasso [51] or Precision Lasso [52] have been developed to reduce instability effects. Usage of such algorithms could be a promising direction of future work on miRGTF-net.

Another potentially valuable application of the proposed method consists in using databases of computationally predicted interactions as opposed to literature-curated databases. Usually, such predictions (e.g., sequence-based prediction of transcription factor binding sites or miRNA targets) contain a large number of false positives which can be further filtered out by miRGTF-net based on mRNA/miRNA expression data. As a proof of concept, we replaced experimentally validated miRNA-mRNA interactions from miRTarBase by computationally predicted pairs (default predictions from TargetScan release 7.2 [53]) and ran miRGTF-net on the same breast cancer data. From more than 700000 interactions only 2216 were present in the constructed network and these entries can be further used to formulate hypotheses on new miRNA-mRNA interactions in breast cancer.

Using genes from the breast cancer interaction network we constructed several classifiers for five-year breast cancer recurrence prediction. As a result, we identified 134 2- to 10-element gene signatures having high AUC, sensitivity and specificity on four independent patient cohorts (two training and two filtration). Almost three-quarters of them (99 classifiers) also showed high prognostic accuracy on an independent validation set. In terms of average quality characteristics, these signatures clearly outperformed previously developed classifiers constructed within the similar exhaustive SVM-based framework [24, 54] using the same microarray datasets. Thus, genes with significant intracellular activity captured by the network construction algorithm were shown to form reliable prognostic signatures.

Fourteen genes were significantly overrepresented in prognostic signatures. This fact highlights their role in breast cancer recurrence. The striking observation consisted in fact that the set of targets of the transcription factor E2F1 significantly intersects with the set of overrepresented genes (ATAD2, CCNB1, FOXM1, KIF2C, PLK1, RACGAP1 and RRM2). Furthermore, both E2F1 and its seven target genes were overexpressed in primary tumors of patients with recurrent cancer. E2F1 belongs to the family of E2F transcription factors and was shown to regulate a large fraction of human genes [55]. Several studies have already mentioned the role of this TF in breast cancer proliferation [56] and metastasis formation [57]. Besides, Vuaroqueaux et. al indicated low expression of E2F1 as a marker of favorable breast cancer outcome [58].

In order to understand processes underlying breast cancer recurrence in tamoxifen-treated patient cohorts, we considered possible interactions and interplay of ESR1 and E2F1. There exists a significant number of genes which can be promoted by both ESR1 and E2F1; in addition, it is known that ESR1 can directly promote E2F1 transcription [43, 44]. The analysis revealed switch of correlation in the groups of patients with and without recurrence. Specifically, ESR1 and E2F1 were not correlated in the group with favorable prognosis, while they were significantly positively correlated in recurrent cancer. These observations together with elevated expression of E2F1 and its prognostic target genes in patients with recurrence, allow

one to hypothesize that ESR1 can be substituted by E2F1 in response to tamoxifen therapy. A similar concept has already been suggested by Miller et al. [59]. In particular, they showed that estrogen receptor alpha can promote an estrogen-independent, E2F-mediated transcriptional program.

Currently the basic platform for transcriptome profiling is RNA sequencing. However the major part of the existing transcriptomic information associated with clinical samples with a sufficient follow-up period has been produced by the microarray technology. Another advantage of network-based gene selection consisted in the ability of cross-platform classifier construction. In particular, we identified the 8-gene signature composed of genes CDK1, FOXM1, LRIG2, MSH2, PLK1, RACGAP1, RRM2 and TMPO, which demonstrated outstanding classification quality on gene expression data obtained by two microarray platforms and RNA sequencing. Specifically, sensitivity and specificity of the constructed signature were both near to 70%. This reliability of the signature is comparable with well-known commercial prognostic test-systems such as OncotypeDX [60] and MammaPrint [61] (according to the report of EGAPP Working Group [62]). Five genes of the signature have already been studied in the context of breast cancer progression and recurrence. Namely, FOXM1 promotes epithelial-mesenchymal transition and metastasis formation for breast and various other cancers [63–65], RRM2 contributes to poor survival and tamoxifen resistance development [66, 67], while expression levels of CDK1, PLK1 and RACGAP1 were shown to be prognostic markers [68–70]. As for LRIG2, MSH2 and TMPO genes, we have not found any evidence on the role of their expression levels in breast cancer pathogenesis. However, existing reports have shown their role in adjacent cancer studies: LRIG2 predicts poor prognosis of uterine cervix carcinoma [71], mutations in MSH2 are associated with Lynch syndrome [72, 73], and TMPO is widely overexpressed in digestive tract and lung cancers [74, 75]. Despite the fact that these genes were not identified as independent markers, each of them brought significant amount of prognostic information to the 8-gene signature. Namely, deletion of LRIG2, MSH2 or TMPO from the signature resulted in significant drop of classification accuracy on several cohorts (even on the training ones)—see S3 Table for details. Notably, six genes of the signature were significantly overrepresented in constructed classifiers while two of them, CDK1 and LRIG2, appeared only in 14 and 19 signatures, respectively.

Among other findings, one of identified strongly connected components in the breast cancer interaction network contained the double-negative feedback loop between miRNAs from mir-200 family and ZEB1/ZEB2 genes. Members of mir-200 family were shown to directly target 3′-UTRs of ZEB1 and ZEB2 which contains eight and nine miRNA binding sites, respectively [76]. At the same time, ZEB1 and ZEB2 can inhibit transcription of these miRNAs by binding to highly conserved sites in their common promoter [77]. Several reports consistently highlight the crucial role of the described loop in cancer metastasis formation [78–80] and epithelial-mesenchymal transition [77, 81, 82]. Our results suggest several other players which can be also involved in this interplay. For example, ZEB1 and ZEB2 repress transcription of miRNAs hsa-miR-107 and hsa-miR-15b-5p which together repress transcription factor FOXO1 (Fig 4). Interestingly, FOXO1 is also present as a hsa-miR-200c-3p direct target [83].

## Materials and methods

### The method for interaction network construction

The first step of a network construction is adding edges from known interaction databases. We used TRRUST v2 [84] for TF–gene interactions, TransmiR v2.0 [85] for TF–miRNA interactions and miRTarBase 7.0 [86] for experimentally validated (reporter assay or western blot)

miRNA–gene interactions. The list of intragenic miRNAs was constructed by the previously described procedure [87].

The next step of the method is incorporation of expression data in established edge structure. First, we discard all weak and wrong-directional interactions. For that we use the following procedure. We calculate Spearman correlation coefficient for each of network edges using miRNA and gene expression data. Then, we remove edges corresponding to interactions of miRNAs and their target genes with positive correlation sign, since we assume that miRNA binding can only down-regulate its target gene. Next, we remove edges corresponding to interactions of host genes and their intragenic miRNAs with negative correlation sign, since we assume that a host gene should be co-expressed with its intragenic miRNAs. Finally, for each interaction type and each regulation direction (positive or negative) we calculate 90% percentile of Spearman correlation absolute value distribution and remove all edges with a module of correlation less than the calculated threshold for corresponding edge type and regulation sign. In other words, we preserve edges labelled with correlation from top-10% of the corresponding group defined by the interaction type and the regulation sign. All self loop edges and isolated nodes are also removed from the network. It is important to note that correlation between two nodes connected by an edge cannot be used as a reliable measure of interaction power, since it ignores the fact that one node can be regulated by many others. Based on this, thresholding absolute values of Spearman correlation is used only as a pre-processing step (edge filtering).

In order to quantify interaction power between two nodes connected by an edge we apply the following procedure. Let $B$ be an arbitrary node and $(A_1, B), \ldots, (A_n, B)$ be all incoming edges. We assume linear dependence of expression values associated with node $B$ (the respective dependent variable is denoted by $y$) on expression values associated with regulator nodes $A_1, \ldots, A_n$ (the respective predictor variables are denoted by $x_1, \ldots, x_n$). That is,

$$y = c_0 + c_1 x_1 + \ldots + c_n x_n + \varepsilon \tag{1}$$

for some real coefficients $c_0, c_1, \ldots, c_n$. We fit the described linear model using the ridge regression technique. The incoming score is then defined as a coefficient of determination $R^2$. This is motivated by the fact that $R^2$ measures the goodness of fit of the considered model ($R^2 = 1$ indicates the ideal fit while $R^2 = 0$ indicates the poorest fit). The interaction score of the $i$th edge $(A_i, B)$ is calculated as the absolute value of the standardized beta coefficient $\beta_i$. This value characterizes the change of $y$ measured in the standard deviation (SD) units under the unit SD change of $x_i$. By default, we remove all nodes not passing a 0.3 threshold on the incoming score to capture noticeable interactions [88]. Edges having interaction score below lower 10% percentile of the respective distribution are also discarded from the network.

## Implementation

The proposed method has been implemented using Python 3 programming language. Stand-alone script, documentation and source codes are available under MIT license at https://github.com/s-a-nersisyan/miRGTF-net. The method inputs gene and miRNA expression data in format of tsv tables containing arbitrary numerical expression estimates in a set of matched samples. The second part of input consists of user-specified miRNA-gene-TF interaction databases. Additionally, arbitrary sign constraints can be set (such as negative correlation of miRNA and target gene expression). Finally, all thresholds used in the algorithm can be varied. For reproducibility, we provided all aforementioned databases as well as TCGA-BRCA expression data as an example. We also provided a manual and a script for importing expression data from arbitrary TCGA projects.

Output of the program includes a report containing summary information on the number of nodes, edges, in-/out-degrees distribution and weakly/strongly connected components. Additionally, the program generates two separate tables for node and edge statistics, including incoming and interaction scores, in- and out-degrees as well as several centrality metrics. Constructed network with its connected components is exported in GraphML format which can be further passed to multiple network analysis and visualization platforms. In the present work we used yED Graph Editor (https://www.yworks.com/products/yed) and Gephi (https://gephi.org).

We also use networkx package [89] for network analysis, scikit-learn [90] for ridge regression fitting, Pandas [91], Numpy [92] and SciPy [93] for miscellaneous computations.

## Breast cancer datasets

The following publicly available RNA expression profiling datasets were utilized:

- mRNA-seq and miRNA-seq data from TCGA-BRCA (data was downloaded from GDC portal https://portal.gdc.cancer.gov/);

- Affymetrix Human Genome U133A Array datasets GSE1456 [94], GSE3494 [95], GSE6532 [96], GSE12093 [97] and GSE17705 [98];

- Affymetrix Human Genome U133 Plus 2.0 Array datasets GSE6532 [96] and GSE20685 [99].

Samples derived from tamoxifen-treated patients with ER-positive tumors were selected for further processing. RNA-seq data was downloaded in a format of expression tables filled with Fragments Per Kilobase Million (FPKM) and reads per million (RPM) values for mRNA-seq and miRNA-seq, respectively. These values were converted into Transcript Per Million (TPM) units and $\log_2$-transformed. Genes and miRNAs having median TPM below 1 were discarded. Microarray data was cleaned from duplicate entries, raw *.CEL files were normalized using RMA algorithm and $\log_2$-transformed using the affy R package [100]. For each gene symbol probeset with maximal median expression across the whole corresponding dataset was taken. The samples were divided into two classes: with recurrence during the first 5 years after surgery or recurrence-free with at least 7 years follow-up. The numbers of patients in each dataset and group are summarized in the S7 Table.

## Classifier construction pipeline

We established a classification pipeline to predict recurrence of ER-positive breast cancer using expression of genes from the network in the primary tumor. At the first step of the pipeline, we sort the list of nodes from the constructed network according to the absolute value of $t$-statistic representing the difference between their expression levels in primary tumors of patients with and without recurrence. Then, we select top-$n$ entries from the obtained list. The procedure used to select the value of $n$ is described below. Additionally, the pipeline has been tested with a gene selection from a complete (not network-specific) list of genes.

After that, we iterate over the set of all possible $k$-element combinations (selection of the value of $k$ is described below) composed of selected genes and measure the prognostic quality of each combination. Namely, for each gene combination we fit a support vector machine (SVM) model on the training set and evaluate it on two filtration sets. In case of successful passage of 0.65 threshold on the minimum of the area under the receiver operating characteristic curve (ROC AUC), sensitivity and specificity for both training and filtration sets classifier is evaluated on the independent validation set. This strategy is similar to the one used in [54].

For Affymetrix Human Genome U133A platform the union of GSE3494 and GSE6532 datasets was used as a training set, GSE12093 and GSE17705 were used as filtration sets and GSE1456 was used as the validation set. For Affymetrix Human Genome U133 Plus 2.0 classifiers were tuned on GSE20685 set and filtered on GSE6532 set. For the TCGA-BRCA dataset, we performed repeated randomized cross-validation with 0.6 threshold value set up on averaged AUC, sensitivity and specificity.

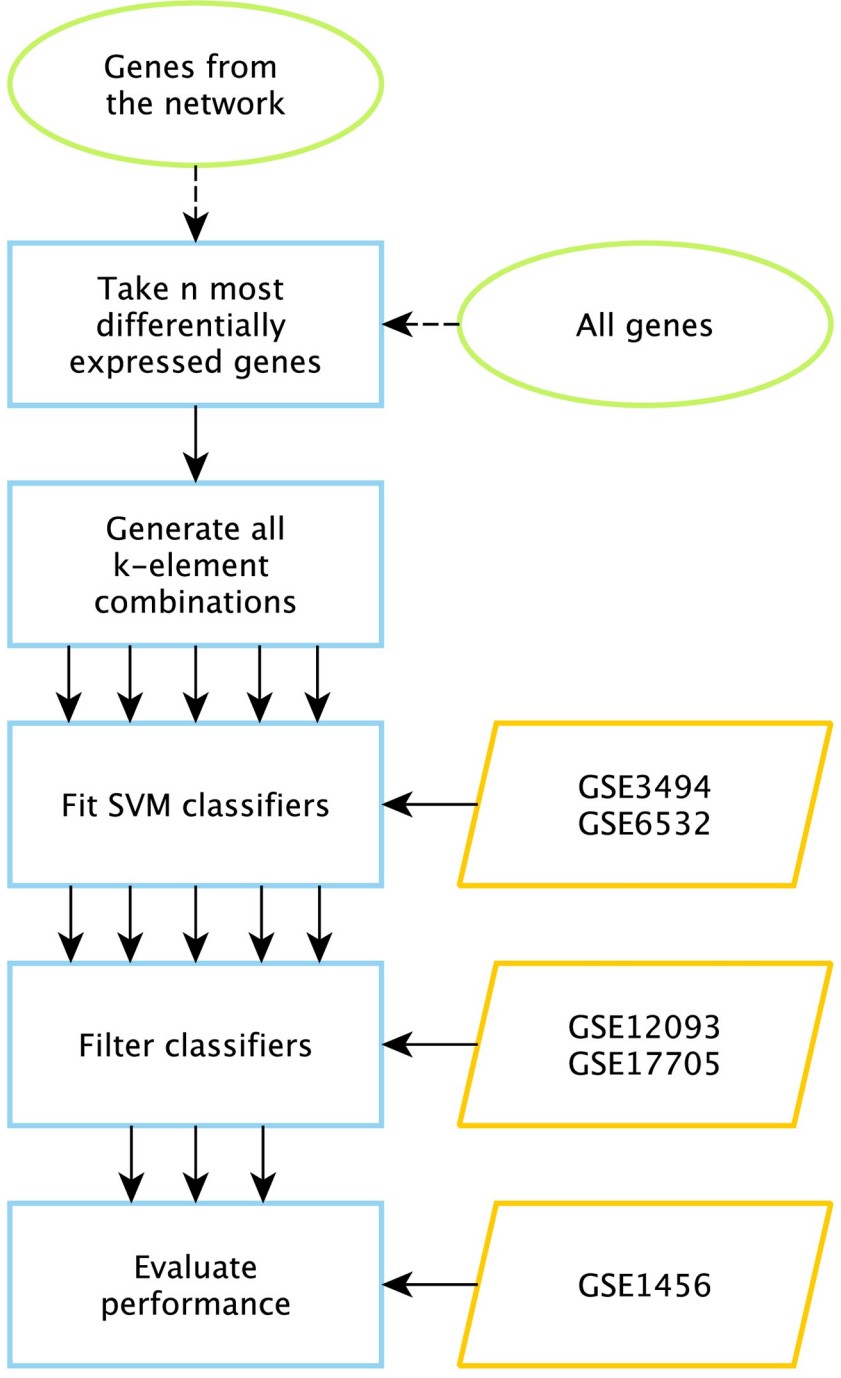

**Fig 7. Classifier construction pipeline.** Green-framed ellipses denote lists of input genes (pipeline can be evaluated in two ways), blue-framed rectangles represent processes, orange-shaped parallelograms represent input data.

We systematically studied the pipeline proposed by executing it for different values of $n$ and $k$ (S8 Table). For each value of $k$ we selected $n$ as the maximum number such that the amount of computations for specified values $n, k$ would not exceed 400 CPU-hours. After that, we incremented $k$ (the procedure started from $k = 1$). The amount of computations was estimated as $O\left(k \times \binom{n}{k}\right)$, since there exist $\binom{n}{k}$ $k$-element combinations for $n$ genes, and SVM model training is linear with respect to dimensionality. The procedure ended when no classifier passed the filtration step for three consecutive values of $k$. General outline of the pipeline is illustrated in Fig 7.

Prior to classification, $z$-score transformation was applied to gene expression data in all sets using mean and variance inferred from the training set. SVM model with the linear kernel and penalty weights inversely proportional to the number of samples in each class was used for classification. Penalty cost parameter $C$ was optimized using random 5-fold stratified cross-validation applied to the training set, balanced accuracy (i.e. sensitivity and specificity weighted by number of samples in each class) was maximized. The following values for $C$ were considered: $4^{-4}, 4^{-3}, \ldots, 4^3, 4^4$. For the TCGA dataset cross-validation procedure was repeated 100 times with different seeds. The LIBSVM wrapper available in scikit-learn package was utilized.

Survival analysis was done using the lifelines Python module (https://zenodo.org/record/4002777#.X06X7Ybgrqo).

### Differential gene expression analysis

Differential expression analysis was performed with Student's $t$-test and DESeq2 [101] for microarray and RNA-seq datasets, respectively.

### Enrichment analysis

Enrichment analysis of gene sets was performed using DAVID v6.8 [102], 0.05 threshold was set on false discovery rates (FDRs) in order to identify significantly enriched terms.

## Supporting information

**S1 Fig. Expression distribution of eight genes from the prognostic signature.** Green color represents patients with recurrence.
(TIF)

**S1 Table. Network genes with the highest out-degrees.**
(XLS)

**S2 Table. Results of DAVID enrichment analysis.**
(XLS)

**S3 Table. Accuracy metrics of prognostic gene signatures.**
(XLS)

**S4 Table. Average performance of gene signatures on the validation set.** $N$ filtration refers to the number of signatures that passed the filtration step, $N$ validation refers to the number of signatures that passed 0.65 threshold on AUC, sensitivity and specificity on the validation set.
(XLS)

**S5 Table. Genes overrepresented in prognostic signatures.**
(XLS)

**S6 Table. Differential expression analysis of E2F1 and its target genes.**
(XLS)

**S7 Table. Number of patients in analyzed cohorts.**
(XLS)

**S8 Table. Values of *n* and *k*.**
(XLS)

## Author Contributions

**Conceptualization:** Stepan Nersisyan, Alexei Galatenko, Vladimir Galatenko, Maxim Shkurnikov, Alexander Tonevitsky.

**Data curation:** Stepan Nersisyan, Alexei Galatenko, Vladimir Galatenko, Maxim Shkurnikov.

**Formal analysis:** Stepan Nersisyan, Alexei Galatenko.

**Investigation:** Stepan Nersisyan, Alexei Galatenko, Vladimir Galatenko, Maxim Shkurnikov, Alexander Tonevitsky.

**Methodology:** Stepan Nersisyan, Alexei Galatenko, Vladimir Galatenko.

**Project administration:** Alexander Tonevitsky.

**Software:** Stepan Nersisyan, Alexei Galatenko.

**Supervision:** Vladimir Galatenko, Maxim Shkurnikov.

**Visualization:** Stepan Nersisyan.

**Writing – original draft:** Stepan Nersisyan, Alexei Galatenko.

**Writing – review & editing:** Stepan Nersisyan, Alexei Galatenko, Vladimir Galatenko, Maxim Shkurnikov, Alexander Tonevitsky.

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
