## [Decision Letter · Decision Letter 0]

25 Feb 2021

PONE-D-20-40860

miRGTF-net: integrative miRNA-gene-TF network analysis reveals key drivers of breast cancer recurrence

PLOS ONE

Dear Dr. Nersisyan,

Thank you for submitting your manuscript to PLOS ONE. After careful consideration, we feel that it has merit but does not fully meet PLOS ONE’s publication criteria as it currently stands. Therefore, we invite you to submit a revised version of the manuscript that addresses the points raised during the review process.

Please check and answer the reviewers' questions and suggestions and provide a revised version of the article and a response to each of the reviewers' points.

We look forward to receiving your revised manuscript.

Kind regards,

Eduardo Andrés-León

Academic Editor

PLOS ONE

Journal Requirements:

Reviewers' comments:

Reviewer's Responses to Questions

**Comments to the Author**

1. Is the manuscript technically sound, and do the data support the conclusions?

Reviewer #1: Yes

Reviewer #2: Yes

2. Has the statistical analysis been performed appropriately and rigorously? 

Reviewer #1: Yes

Reviewer #2: Yes

3. Have the authors made all data underlying the findings in their manuscript fully available?

Reviewer #1: Yes

Reviewer #2: Yes

4. Is the manuscript presented in an intelligible fashion and written in standard English?

Reviewer #1: Yes

Reviewer #2: Yes

5. Review Comments to the Author

Reviewer #1: Comments to the author:

The authors report miRGTF-net, a tool for constructing regulatory networks among miRNAs, genes, and TFs. This tool combines a database-level approach as well as expression profiles given from miRNAs and genes to reduce potential false positives for edge construction. They have applied miRGTF-net on the TCGA-BRCA ER-positive dataset and proposed a classification pipeline to predict recurrence from several independent patient cohorts. The presentation is in a light quality level and in the scope of PLOS ONE. The rationale of the work is interesting, although it’s not completely novel. The implementation can be considered of some significance for the research community. Several concerns should be addressed as stated below that I believe would increase the quality and clarity of the manuscript.

Major comments:

1. P2 L28-33: The two major approaches of network construction are described their respective disadvantages but no advantage is mentioned here. I suggest using “limitations” in the sentence rather than advantages and disadvantages.

2. P2 L29: “disadvantagesr” should be “disadvantages”.

3. P2 L29-30: Why does the database-level analysis usually lack tissue specificity? Please describe it in detail.

4. P3 L82-83: “some other node” should be “some other nodes”.

5. P4 L118: How many genes were used as the input for the enrichment analysis? What is the threshold used here to identify the significantly enriched terms? Please add these in the article. Also, check out the number of 218 significantly enriched terms is correct.

6. P5 L156: Add a comma between “potential” and “we”.

7. P8 L276: “intersect” should be “intersects”.

8. P9 L341-342: Are miRNA IDs unified to the latest version of miRBase when constructing the miRGTF-net databases? Among the databases, miRTarBase 7.0 and TransmiR v2.0 utilize miRBase v21 and v22, respectively. According to the description of the miRNA.diff file from miRBase, there are 12 human mature miRNAs have been changed the IDs between the two versions. Although the impact is slight, still might cause the relationship in a network to unmatch. Additionally, the miRGTF-net input data with miRNA IDs can use an annotation belonging to one of the various versions of miRBase. Once these miRNA IDs use older versions, the number of miRNAs that cannot be mapped to the databases correctly will increase. This case is more common in early miRNA-seq data. The authors should address this issue and propose a feasible approach in miRGTF-net.

9. Figs. 1 and 7: Please use standard flowchart symbols instead of colors to represent elements in the workflows.

Comments for the miRGTF-net improvement:

1. Besides the human databases, the authors can consider establishing the databases of other model organisms for miRGTF-net.

2. It would be good to show the real-time progress/status as well as brief summary while running the miRGTF-net script.

Reviewer #2: In this manuscript, Nerissyan et al. introduced a novel method to analyze miRNA-gene-TF network on breast cancer datasets. This method highlights the use of integrative information (microarray data, TCGA database, miRNA information) to build a classifier for five-year breast cancer recurrence prediction. As a result, they have identified a set of gene signature that can be used to predict the outcome of cancer patients. Finally, the authors discussed the mechanistic roles of ESR1 and E2F1 underlying breast cancer recurrence.

However, I think the following points need to be clarified in the manuscript:

1, Since the results are based on the input databases, and the use of databases may bias the output. I would like to see a comparison between using other miRNA prediction tools, such as TargetScan, to perform the analysis and see how robust the results would be.

2, I can see the potential of applying this software to other species. I suggest the author to include a note on the Github pages and show how to build the input data for other model organisms, such as mouse and Drosophila. If the preparation of those data needs pre-processing, they should also include scripts to perform that.

3, I suggest the authors to compare one or two more machine learning methods in the classification step. Although SVM is considered as a good classifier (AUC value), other algorithm may outperform it due to the use of different strategy, such as tree models (decision tree, random forest), and probability-based method, for example, Naive Bayes classifier.

4, I suggest the authors to provide more analysis/visualization on the gene signatures, for example, how stable are the signature across TCGA samples? Do they have individual variations? A Heatmap with clustering analysis would be a useful way to address these questions.

Also, I have tested the scripts on their GitHub webpage and the software is user friendly. I would suggest the authors to change the default output format to pdf, which is easier to open and process than the current graphml format. Overall, this work provides a new angle for integrative TF and miRNA analysis. I suggest a moderate revision of the manuscript before publication.

6. PLOS authors have the option to publish the peer review history of their article (what does this mean?). If published, this will include your full peer review and any attached files.

Reviewer #1: No

Reviewer #2: **Yes: **Yu H. Sun

---

## [Author Response · Author response to Decision Letter 0]

8 Mar 2021

Reviewer 1

Point 1. P2 L28-33: The two major approaches of network construction are described their respective disadvantages but no advantage is mentioned here. I suggest using “limitations” in the sentence rather than advantages and disadvantages.

Response 1. We thank the Reviewer for pointing this out. We revised the text accordingly.

Point 2. P2 L29: “disadvantagesr” should be “disadvantages”.

Response 2. This word was removed from the text (see Response 1).

Point 3. P2 L29-30: Why does the database-level analysis usually lack tissue specificity? Please describe it in detail.

Response 3. The detailed explanation was added to the paragraph.

Point 4. P3 L82-83: “some other node” should be “some other nodes”.

Response 4. The typo was corrected.

Point 5. P4 L118: How many genes were used as the input for the enrichment analysis? What is the threshold used here to identify the significantly enriched terms? Please add these in the article. Also, check out the number of 218 significantly enriched terms is correct.

Response 5. The required information was added to the manuscript text (lines 122, 479-480). We also verified that there are 218 significantly enriched terms (the last column of S2 Table with 0.05 threshold was used).

Point 6. P5 L156: Add a comma between “potential” and “we”.

Response 6. Comma was added.

Point 7. P8 L276: “intersect” should be “intersects”.

Response 7. The typo was corrected.

Point 8. P9 L341-342: Are miRNA IDs unified to the latest version of miRBase when constructing the miRGTF-net databases? Among the databases, miRTarBase 7.0 and TransmiR v2.0 utilize miRBase v21 and v22, respectively. According to the description of the miRNA.diff file from miRBase, there are 12 human mature miRNAs have been changed the IDs between the two versions. Although the impact is slight, still might cause the relationship in a network to unmatch. Additionally, the miRGTF-net input data with miRNA IDs can use an annotation belonging to one of the various versions of miRBase. Once these miRNA IDs use older versions, the number of miRNAs that cannot be mapped to the databases correctly will increase. This case is more common in early miRNA-seq data. The authors should address this issue and propose a feasible approach in miRGTF-net.

Response 8. We corrected errors related to miRBase v21 to v22 migration, which resulted in only one new interaction between miRNA and its target gene, and one additional two-node strongly connected component. This interaction, however, had not modified the major weakly connected component and, therefore, constructed classifiers remained unchanged. We updated the corresponding part of the manuscript (lines 104, 132-133). Additionally, we uploaded miRTarBase version 8.0 to the miRGTF-net repository as a default database for miRNA-gene interactions and provided the link for miRBase name conversion tool (miRBaseConverter).

Point 9. Figs. 1 and 7: Please use standard flowchart symbols instead of colors to represent elements in the workflows.

Response 9. Figures 1 and 7 were updated accordingly.

Point 10. Besides the human databases, the authors can consider establishing the databases of other model organisms for miRGTF-net.

Response 10. We added three mouse databases to the miRGTF-net GitHub repository (TRRUST, TransmiR, miRTarBase), so one can run the tool if miRNA/mRNA expression data in a set of samples are available. This information was also added to the manuscript text (lines 97-98).

Point 11. It would be good to show the real-time progress/status as well as brief summary while running the miRGTF-net script.

Response 11. Progress messages and a brief summary were added to the run.py script.

Reviewer 2

Point 1. Since the results are based on the input databases, and the use of databases may bias the output. I would like to see a comparison between using other miRNA prediction tools, such as TargetScan, to perform the analysis and see how robust the results would be.

Response 1. We thank the Reviewer for pointing this out. Since TargetScan is a tool for in silico sequence-based target prediction, it contains about 25 times more interactions compared to miRTarBase subset for interactions with strong experimental support. Thus, a direct comparison of two networks is inapplicable. However, use of TargetScan in miRGTF-net can allow one to formulate hypotheses on new miRNA-mRNA interactions in a considered tissue type. We expanded the Discussion section accordingly (lines 269-279).

Point 2. I can see the potential of applying this software to other species. I suggest the author to include a note on the Github pages and show how to build the input data for other model organisms, such as mouse and Drosophila. If the preparation of those data needs pre-processing, they should also include scripts to perform that.

Response 2. We added three mouse databases to the miRGTF-net GitHub repository (TRRUST, TransmiR, miRTarBase), so one can run the tool if miRNA/mRNA expression data in a set of samples are available. This information was also added to the manuscript text (lines 97-98).

Point 3. I suggest the authors to compare one or two more machine learning methods in the classification step. Although SVM is considered as a good classifier (AUC value), other algorithm may outperform it due to the use of different strategy, such as tree models (decision tree, random forest), and probability-based method, for example, Naive Bayes classifier.

Response 3. Aside from the SVM classifier, we also tried several alternatives such as mentioned random forests, Naive Bayes as well as others (k-nearest neighbors, gradient boosting). Interestingly, all these methods had not allowed us to construct classifiers with reliable quality, so we decided not to include this information into the text, since this could shift accents of the manuscript far from the main line (network construction and analysis).

Point 4. I suggest the authors to provide more analysis/visualization on the gene signatures, for example, how stable are the signature across TCGA samples? Do they have individual variations? A Heatmap with clustering analysis would be a useful way to address these questions.

Response 4. Heatmaps with clustering for each dataset were added (S1 Figure, lines 209-210).

Point 5. I would suggest the authors to change the default output format to pdf, which is easier to open and process than the current graphml format.

Response 5. We thank the reviewer for the suggestion for improving the tool. Unfortunately, we were not able to adequately address the problem of universal programmatic graph visualization yet, but we are planning to work on this feature for the future releases of miRGTF-net. However, we added links to Gephi and yED Graph Editor to the GitHub README file, so one can simply use these tools for fast and beautiful visualization of networks from graphml format.

---

## [Decision Letter · Decision Letter 1]

18 Mar 2021

miRGTF-net: integrative miRNA-gene-TF network analysis reveals key drivers of breast cancer recurrence

PONE-D-20-40860R1

Dear Dr. Nersisyan,

We’re pleased to inform you that your manuscript has been judged scientifically suitable for publication and will be formally accepted for publication once it meets all outstanding technical requirements.

Kind regards,

Eduardo Andrés-León

Academic Editor

PLOS ONE

Reviewers' comments:

Reviewer's Responses to Questions

**Comments to the Author**

1. If the authors have adequately addressed your comments raised in a previous round of review and you feel that this manuscript is now acceptable for publication, you may indicate that here to bypass the “Comments to the Author” section, enter your conflict of interest statement in the “Confidential to Editor” section, and submit your "Accept" recommendation.

Reviewer #1: All comments have been addressed

Reviewer #2: All comments have been addressed

2. Is the manuscript technically sound, and do the data support the conclusions?

Reviewer #1: Yes

Reviewer #2: Yes

3. Has the statistical analysis been performed appropriately and rigorously? 

Reviewer #1: Yes

Reviewer #2: Yes

4. Have the authors made all data underlying the findings in their manuscript fully available?

Reviewer #1: Yes

Reviewer #2: Yes

5. Is the manuscript presented in an intelligible fashion and written in standard English?

Reviewer #1: Yes

Reviewer #2: Yes

6. Review Comments to the Author

Reviewer #1: In this revised version, the authors exhaustively addressed all the raised concerns. I consider these changes made by the authors to be appropriate. The manuscript is overall more curated, therefore, I suggest the publication of the paper in its revised form.

Reviewer #2: In this revised article, the authors have addressed all the questions I raised during the first round of review. In addition, they added more discussions in the main text (lines 269-279) which expands the application of the new method they developed. The updated github files (such as TRRUST, TransmiR, miRTarBase databases, and the instructions of visualization) will also benefit the users to implement their software in their own research. I hope that the authors keep good maintenance of the github repository and regularly check the reported bugs raised by the users. I suggest to accept this manuscript.

7. PLOS authors have the option to publish the peer review history of their article (what does this mean?). If published, this will include your full peer review and any attached files.

Reviewer #1: No

Reviewer #2: **Yes: **Yu H. Sun

---

## [Editor Report · Acceptance letter]

19 Mar 2021

PONE-D-20-40860R1 

miRGTF-net: integrative miRNA-gene-TF network analysis reveals key drivers of breast cancer recurrence 

Dear Dr. Nersisyan:

I'm pleased to inform you that your manuscript has been deemed suitable for publication in PLOS ONE. Congratulations! Your manuscript is now with our production department. 

Kind regards, 

on behalf of

Dr. Eduardo Andrés-León 

Academic Editor

PLOS ONE